# Evaluation of Drinking Water Quality and Bacterial Antibiotic Sensitivity in Wells and Standpipes at Household Water Points in Freetown, Sierra Leone

**DOI:** 10.3390/ijerph19116650

**Published:** 2022-05-29

**Authors:** Dauda Kamara, Doris Bah, Momodu Sesay, Anna Maruta, Bockarie Pompey Sesay, Bobson Derrick Fofanah, Ibrahim Franklyn Kamara, Joseph Sam Kanu, Sulaiman Lakoh, Bailah Molleh, Jamie Guth, Karuna D. Sagili, Simon Tavernor, Ewan Wilkinson

**Affiliations:** 1Water Sanitation and Hygiene (WASH) Program, Ministry of Health and Sanitation, Freetown 00232, Sierra Leone; 2Directorate of Environmental Health & Sanitation, Ministry of Health and Sanitation, Freetown 00232, Sierra Leone; dbah@mohs.gov.sl (D.B.); sesaydu59@yahoo.com (M.S.); 3World Health Organization (WHO) Country Office, Freetown 00232, Sierra Leone; marutaa@who.int (A.M.); sesaybo@who.int (B.P.S.); fofanahb@who.int (B.D.F.); ikamara@who.int (I.F.K.); 4National Disease Surveillance Program, Ministry of Health and Sanitation, Sierra Leone National Public Health Emergency Operations Centre, Freetown 00232, Sierra Leone; samjokanu@yahoo.com; 5Department of Medicine, University of Sierra Leone Teaching Hospitals Complex, Freetown 00232, Sierra Leone; 6Department of Medicine, College of Medicine and Allied Health Sciences, University of Sierra Leone, Freetown 00232, Sierra Leone; lakoh2009@gmail.com; 7Department of Community Health, Faculty of Clinical Sciences, College of Medicine and Allied Health Sciences, University of Sierra Leone, Freetown 00232, Sierra Leone; 8Sustainable Health Systems Sierra Leone, 34 Military Research Center, Freetown 00232, Sierra Leone; bmollehshs@gmail.com; 9Global Health Connections, Center Barnstead, Barnstead, NH 03225, USA; guth.jamie@gmail.com; 10International Union Against Tuberculosis and Lung Disease (The Union), South East Asia Office, New Delhi 110016, India; ksagili@theunion.org; 11School of Medicine, University of Liverpool, Liverpool L69 3GE, UK; s.j.tavernor@liverpool.ac.uk; 12Institute of Medicine, University of Chester, Chester CH2 1BR, UK; e.wilkinson@chester.ac.uk

**Keywords:** clean water, standpipes, community wells, water contamination, nitrates, public water company, Structured Operational Research Training Initiative (SORT IT), operational research

## Abstract

Water quality surveillance can help to reduce waterborne diseases. Despite better access to safe drinking water in Sierra Leone, about a third of the population (3 million people) drink water from unimproved sources. In this cross-sectional study, we collected water samples from 15 standpipes and 5 wells and measured the physicochemical and bacteriological water quality, and the antimicrobial sensitivity of *Escherichia coli* (*E. coli*) in two communities in Freetown, Sierra Leone in the dry and wet seasons in 2021. All water sources were contaminated with *E. coli*, and all five wells and 25% of standpipes had at least an intermediate risk level of *E. coli*. There was no antimicrobial resistance detected in the *E. coli* tested. The nitrate level exceeded the WHO’s recommended standard (>10 parts per million) in 60% of the wells and in less than 20% of the standpipes. The proportion of samples from standpipes with high levels of total dissolved solids (>10 Nephelometric Turbidity Units) was much higher in the rainy season (73% vs. 7%). The level of water contamination is concerning. We suggest options to reduce *E. coli* contamination. Further research is required to identify where contamination of the water in standpipes is occurring.

## 1. Introduction

Safe drinking water, and good sanitation, are important to prevent the spread of waterborne diseases [1]. Safe drinking water is defined as water that does not represent any significant risk to health over a lifetime of consumption including different sensitivities that may occur between life stages [2]. It is estimated that, globally, 1.4 million deaths (10%) of children under the age of five years are due to unsafe drinking water. About 9% of the global burden of disease and 6% of all deaths are due to unsafe water, inadequate sanitation, and poor hygiene [3], which together cause diseases such as cholera, diarrhea, dysentery, hepatitis A, typhoid and malnutrition [3]. Globally, diarrhea is reported to kill more children than malaria and tuberculosis combined [3].

Nearly 2 billion people in low- and middle-income countries (LMICs) use a source of drinking water contaminated with feces [3]. Considering clean drinking water to be a fundamental human right, the United Nations prioritized it in Sustainable Development Goal (SDG) 6 “availability and sustainable management of Water, Sanitation and Hygiene (WASH) for all” [4]. Several interventions were taken up by governments and development agencies to reduce water contamination, in particular from contamination with fecal matter, by nature of construction or through active intervention [5]. These are commonly piped water supplies, wells with watertight sheaths, which reduce contamination by bacteria from the surrounding soil, or springs protected from contamination. However, even in improved sources, there are reports of fecal contamination with up to 25% of samples being contaminated with *Escherichia coli* (*E. coli*) [6]. The World Health Organization (WHO) recommends that water with any fecal coliforms should not be consumed [2]. Water quality is therefore critical to ensure public health, making safe drinking water one of the important public health priorities [7].

In sub-Saharan Africa, 76% of the population does not have access to safe drinking water reflecting significant socio-economic disparities across different countries, whether rich or poor, urban or rural, and within slum communities [8]. The incidence of diarrheal disease in rural areas (67%) is much higher than that in urban areas (33%), which reflects the lack of improved water and sanitation facilities in rural areas [9]. It is not just a problem with water being contaminated when collected. Studies from rural South Africa reported that households storing water for human consumption were at high risk of diarrheal diseases [10] and a study in Nigeria among children under five showed the impact of household risk factors related to the severity of diarrheal diseases [8].

Exposure to microbial pathogens through the contaminated water supplies is a particular problem in low-income countries, causing a significant proportion of the public health disease burden [11]. Pathogens such as bacteria, viruses, and parasites commonly occur in contaminated water. *E. coli* are bacteria found in the environment, foods, and intestines of people and animals. Most strains of *E. coli* can make people and animals ill by causing diarrhea, urinary tract infections, respiratory illness and pneumonia, and other illnesses [12]. Overall, the most common contaminant of water sources is human excreta, particularly fecal pathogens, and parasites [7].

*E. coli* is a widely accepted indicator organism for assessing contamination of drinking water [13] and is the only true fecal coliform. It is a reliable indicator of human or animal fecal contamination. It has the advantage of there being a number of sensitive, specific, inexpensive, easy-to-use methods available for its detection directly from water samples [14]. Guidelines for drinking water quality recommend that *E. coli* should not be detected in any 100 mL of drinking water sample [2].

The consumption of antibiotics in both human treatment and agricultural processes has increased greatly [15]. Recently, antibiotic resistance has become a major public health issue [16,17,18] and the presence of resistant organisms in wastewater, surface water, and drinking water is well documented and is of concern [17,18,19,20]. For example, high antibiotic resistance rates of 59% and 72% were found in a drinking water source in Guinea-Bissau (West Africa) during the dry and wet seasons [21]. High antibiotic resistance levels were also observed in the drinking water sources of the Huangpu River of China [22]. Studies in South Africa have shown that *E. coli* in tap water was resistant to a number of common antibiotics, reflecting their antimicrobial use in medicine and in animal husbandry [23]. Since antibiotic-resistant bacteria have an impact on the effectiveness of clinical use of antibiotics for the management of infectious disease, environment surveillance is critical [24].

Apart from pathogens contaminating drinking water, water’s physicochemical aspects are also important. These may affect both the acceptability of the water to the consumer and the practicality of water treatment. The hydrogen ion concentration (pH) should be less than 8.0 and turbidity should ideally be below 0.1 Nephelometric Turbidity Units (NTU) [2].

Chemical contaminants such as nitrates, phosphates and lead may be found in water at levels that are detrimental to health. Nitrates are widely used in fertilizer, which may then be washed into water sources. Levels of over 10 parts per million may cause methemoglobinemia in babies and can be associated with some cancers and birth defects [25]. Lead may be dissolved in drinking water. It can accumulate in the body, resulting in lead poisoning, which may cause anemia as the lead interferes with the formation of hemoglobin by preventing iron uptake. Higher levels of lead in the blood may result in permanent brain damage and kidney dysfunction [26]. Zinc toxicity in humans may rarely occur [26] but the WHO guidelines for drinking water quality do not set a recommended limit as exposure to high levels is unlikely as the water is unpalatable [2]. Phosphate may be added as a corrosion inhibitor in drinking water distribution systems to provide insoluble phosphate scales which coat the pipe walls. This prevents iron and lead corrosion products entering the water supply, and thus prevent lead poisoning [27]. In Sierra Leone, about 3 million people still drink water from unimproved sources, although this is reducing with increased access to basic water sources (meaning improved drinking water sources such as pipe water systems, and protected wells, boreholes, and springs), having increased from 40% in 2000 to 61% in 2017 [8]. In 2016, there were 31,345 water points according to the WASH survey and water point mapping [28]. However, out of the improved water sources available then, around 30% were not functional at a specific period or unavailable for use due to minor or major damage. Additionally, only 11% of households were reported to use drinking water sources that were free from fecal contamination [8], of which only 3% have drinking water free from fecal contamination at the point of use. Interventions are being carried out by the WASH program within the Ministry of Health and Sanitation (MoHS) to improve the quality of household water and safe storage as indicated in their annual WASH report [29]. One of the key mandates of the Ministry of Water Resources (MoWR), as embedded in the national strategy of the Water Safety Plan (WSP), is to oversee the quality of the public water supply.

In Freetown, the majority of households rely on water supplied by the public water company to community standpipes, while the affluent members of the city have the public water company’s water piped directly into their houses. Standpipes are either directly connected to the piped network from the water company, or are attached to a 10,000 L water tank which is replenished by water bowsers. Once water enters the distribution system, it should remain uncontaminated as it makes its way from the treatment plant to the final consumer. If pipes are not well maintained, or there are poor handling practices during transportation, the water may be contaminated after leaving the treatment plant [8].

Some people access water from unprotected water sources, such as hand dug wells which are not lined, because they may live far away from standpipes or water is not available in the standpipes during the dry season (November–April) [8].

A literature search conducted in January 2022 yielded no studies from Sierra Leone that assessed water quality, including antimicrobial resistance (AMR). The only data available were from the 2017 Multiple Indicator Cluster Survey (MICS6), which included some data on water quality as part of monitoring progress towards the Sustainable Development Goals (SDGs) and other internationally agreed commitments [8].

This study was designed to review water quality, including antibiotic resistance, within two communities in Freetown, which access water through standpipes and wells.

## 2. Materials and Methods

### 2.1. Study Design

This was a cross-sectional study involving primary drinking water sample collection.

### 2.2. General Setting

Sierra Leone is a country on the Atlantic west coast of Africa. It is bordered by Liberia on the southeast and Guinea on the northeast. It has an estimated population of 8 million with sixteen (16) districts. Freetown is the capital and the largest city and the Western Area is the most populated area (1,055,964) [30]. Sierra Leone has two main seasons—the dry season (November to April) and the rainy/wet season (May to October). The highest amount of rainfall is experienced in August and the lowest in April.

There is one public water company that supplies the piped water in Freetown through 176 miles of pipes. The system was initially designed to serve a population of half a million people [31], but Freetown now has a population of more than a million. The water network is estimated to reach 60% of the population in Freetown [32]. In 2017, the mandate of the company was expanded to cover the whole of the Western Area with a population of 1.5 million residents.

The public water company relies principally on a single source dam, with over 90% of the total water supplied to Freetown coming from the dam and the water treatment plant at Mile 13. However, there are other secondary water sources (coming from the Kongo Dam, Sugar Loaf, Charlotte Weir, Blue Cemetery, and White Water), which are seasonal and only contribute to the water supply during the rainy season and early in the dry season. The water from these secondary sources is often channeled to the main dam. Deforestation activities are an appreciable problem and are leading to a reduction in water collected in the catchment areas [32].

There are a number of private bottling and sachet water companies in Sierra Leone selling drinking water to people. However, access to this water is dependent on people’s purchasing capacity.

The flow diagram of the public water company (see Figure 1) depicts the water traveling from the water catchment area to the sedimentation tank, and then onward to the chlorination tank, the final treatment tank, the storage tank or reservoir, and finally through the distribution networks either to standpipes connected to the water company’s pipe network or by transportation via water bowser to the storage tank with standpipes connected to storage tanks of 10,000 L capacity.

### 2.3. Site Specific Setting

The Brookfields and Wilberforce communities are situated in the capital city, Freetown. Brookfields is a community of 62,499 living near central Freetown, with a mixture of residential and commercial properties, government ministries and informal agricultural activity. Males constitute 51% and females 49% [30].

Wilberforce is situated on the hills above central Freetown, with mainly residential housing, a few commercial buildings, a large military barracks and the national military referral hospital. The area’s population is 53,981, encompassing 49% males and 51% females, and is more affluent than Brookfields [30]. The water quality sampling and inspections were carried out in the dry season before the rains started in May 2021 and when the rainfall was high in the rainy season in August 2021.

We inspected the physical condition of, and took water samples from, five wells and fifteen standpipes: three wells and seven standpipes in Brookfields and two wells and eight standpipes in Wilberforce as shown in Figure 2. Standpipes and wells were randomly selected from these two locations from a list of water points given by the MoWR.

### 2.4. Study Subjects

The study subjects were water sources in community wells, community standpipes and the water reservoir in the two communities (Brookfields and Wilberforce) in Freetown.

### 2.5. Collection of Samples

Water samples were collected over a period of five days in early May (before the start of the rains) and in August 2021 (the wet season) by the corresponding author, two co-authors and two laboratory analysts. Six data collectors were also part of the sample collection team (three each from Wilberforce and Brookfields). The data collectors were oriented on the purpose of the study and necessary precautions to be taken to prevent sample contamination during the collection.

To measure the water quality delivered from the water company through the standpipes, water was sampled at four points as the water passed progressively through the water treatment plant, the final sample being taken at the outflow after purification treatment. Samples were then taken from the 15 standpipes: eight in Brookfields and seven in Wilberforce. Water was also sampled from the five community wells in both Brookfields and Wilberforce. This gave a total of 48 samples that were collected, 24 in each season. Distilled water was used to clean the pipette, thermometer, pH meter and turbidity meter between measurements. Collected water samples were transported on an icepack to the Milton Margai Technical University Laboratory (MMTU) for laboratory analysis.

### 2.6. Laboratory Processing of Samples

In the laboratory, a pH meter, turbidity meter and spectrophotometers were used to test for the physicochemical parameters, measuring pH, turbidity (Total Dissolved Solids: TDS), and levels of nitrate, lead, zinc and phosphate [2].

#### 2.6.1. Physical Assay

On-site measurements of temperature, pH and TDS were conducted using a mercury-in-glass thermometer, digital pH and turbidity meter.

Conductivity, salinity, and TDS were measured using the HACH potable conductivity meter (CO150). Turbidity is measured in Nephelometric Turbidity Units (NTU). The standard range of turbidity used and effective for disinfection of drinking water were below 5NTU, ideally less than 1NTU. Turbid drinking water recording greater than 10NTU should not be chlorinated unless pre-treatment has been performed [2].

#### 2.6.2. Chemical Assay

Nitrate (NO^3−^), lead (Pb), zinc (Zn) and phosphate or orthophosphate ion [PO4]^3−^ determined spectrophotometrically [33].

##### Nitrate

The materials used were a UV–VIS spectrophotometer, volumetric flasks (100 mL), distilled water, number 41 filter paper, beakers, pipettes, nitrate stock solution, concentrated hydrochloric acid, 0.5% sulfanilic acid, 0.5% methyl anthranilate, and sodium hydroxide solution. This was placed into a beaker, including 10 mL of nitrate stock solution, 5 mL of concentrated hydrochloric acid and 2 mL Sodium Chloride (NaCl) granular mixture were pipetted and they were left to stand for 30 min with occasional stirring to form nitrate.

The solution was filtered into a 100 mL standard flask and diluted up to the mark. A liquid stock solution containing 0.26–10.7 mg/mL of reduced nitrate was transferred into a series of 10 mL standard flasks. Then, 1 mL of 0.5% sulfanilic acid and 1 mL of 2 m hydrochloric solution were thoroughly shaken for 5 min for the diazotization reaction to occur. Nitrate absorbance was determined spectrophotometrically at 410 nm. The various nitrate concentrations in parts per million (ppm) were obtained from the water samples [34,35].

##### Lead

A UV–VIS spectrophotometer, pH meter, furan-2-Carbaldhyde, acetic acid, ethyl alcohol (90%), dimethyl sulfoxide, lead acetate, benzene-1-3-amine, and deionized water were used to perform this test. The preparation of reagents involved four processes: (1) Ligand reagents: stock solution (100 mL) of ligand reagent of concentration (1 × 10^−2^ m) was prepared by dissolving 0.264 g of ligand reagent in the minimum amount of dimethyl sulfoxide to serve as a solvent and placed filled in 100 mL of deionized water. (2) Standard lead solution: lead (ii) stock solution of 0.001 m was prepared by dissolving 0.0325 g of Pb(C_2_H_3_O_2_)_2_ in 100 mL of deionized water and solutions of lower concentrations were prepared from the stock solution by dilution; the solution was standardized by EDTA. (3) Surfactant solutions: the surfactants—sodium laureth sulfate, triton × 100 and cetyl trimethyl ammonium bromide—were prepared at concentration (1 × 10^−2^ m) by dissolving in deionized water and used without further purification. (4) Buffer solutions: for the preparation of buffer solutions, different types of buffers including acetate buffer in the range (3.5–6), universal buffer (1.81–11.98), borate buffer (7.6–10), citrate buffer, phosphate buffer and acetate buffer (3.5–6) were prepared.

A mixture of substituted benzene-1, 3-amine (10 millimole (mmol)) (1.08 g) and furan-2-carbaldhyde (20 mmol) (1.65 mL) was dissolved in 10 mL absolute ethanol, and then was refluxed for 5 h. It was cooled and diluted with ice cold water. The resulting solid was recrystallized from ethanol to Schiff bases.

The complex (solid) was prepared by addition of 10 mL hot solution (0.1 m) lead acetate to 20 mL hot solution (40 degree Celsius) of the Schiff’s base BDFM (0.1 m) with 3 mL of acetate buffer pH 3.5. The resulting mixture was stirred under reflux for 1 h. The samples were then analyzed spectrophotometrically using a spectrophotometer at 283.306 nanometers (nm) [34].

##### Zinc

The materials used were 3-hydroxybenzylaminobenzoic acid, chemical balance, a pH meter, an Ultraviolet spectrophotometer, 3-hydroxylbenzaldehyde, 4-aminobenzoic acid, ethanol, volumetric flasks, distilled water and n-butanol.

One gram of 3-hydroxylbenzaldehyde was dissolved in 25 mL of double distilled water, mixed in a flask with 1 g of 4-aminobenzoic acid and refluxed for 3 h.

A pale-yellow colored crystal product was formed. After filtration, it was dried at room temperature.

The product was recrystallized using ethanol. Zinc was reacted with 3-hydroxybenzylaminobenzoic acid to form a light-yellow-colored complex at pH 5.0. One mL of zinc (ii) solution was transferred into a 25 mL standard flask and to 3 mL of buffer (pH 5.0), 2 mL of 3-hydroxylbenzylaminobenzoic acid solution was added. Its volume was brought up to 10 mL with double distilled water.

The absorption spectrum of the reagent was recorded. The reagent metal complex gave maximum absorbance at 620 nm (wavelength). The pH values of aqueous phases obtained varied buffer solutions ranging between 2.0 and 8.0. The volume of each aqueous phase was adjusted to 10.0 mL with double distilled water.

The concentration of 3-hydroxylbenzylaminobenzoic acid varied between 1.0 × 10^−3^ m and 11.0 × 10^−3^ m and displayed maximum color formation. The total volume of aqueous phase was brought to 10.0 mL with double distilled water. The aqueous phases were stirred with 10.0 mL of n-butanol; the organic phases were collected in 25 mL standard flasks and made up to 25 mL with n-butanol. The zinc absorbance of these phases was determined at 620 nm, against their corresponding reagent blanks using the spectrophotometer [34].

##### Phosphate Analysis

For phosphate determination, 5 N sulphuric acid, potassium antimonyl tartrate, ammonium molybdate, sodium phosphate monobasic monohydrate, deionized water, a spectrophotometer and ascorbic acid were utilized.

Fifty milliliters of 5 N sulfuric acid, 5 mL of potassium antimonyl tartrate, 15 mL ammonium molybdate, and 30 mL ascorbic acid were prepared. In summary, six standards were then prepared, each containing 4 mL of the combined reagent and 25 mL of sodium phosphate monobasic monohydrate.

Each solution was allowed to stand for ten minutes. The absorbance of the samples was measured using a UV–VIS spectrophotometer at a wavelength of 840 nm, with the results recorded. The combined reagents were disposed of, and the phosphorous standards were formed [34,35].

#### 2.6.3. Bacteriological Analysis

Bacteriological water quality analysis was carried out on all the water samples using the Most Probable Number (MPN) method.

A sterilized conical flask with 40 g of MacConkey Broth was added to 1000 mL of distilled water which provided a single strength medium.

In another flask, 80 g of the MacConkey broth in 1000 mL of distilled water produced a double strength medium.

The composition of MacConkey broth requires 20 g of peptic digest tissue, 10 g of lactose, 5 g of bile salt, 0.75 g of neutral red and 1 L of distilled water. Then, the flasks were sterilized in an autoclave for 15 min at 121 °C. After autoclaving, the water samples were placed in a series of test tubes. The known concentration was diluted with water in the three sets of tubes: 10 mL in the first tube, 1 mL in the other tube and 0.1 mL in the last tube.

Ten milliliters of double strength MacConkey medium was added to 10 mL of water sample, while 10 mL of single strength MacConkey medium was added to 1 mL and 0.1 mL water sample, respectively.

We incubated the tubes for 24 to 48 h at 35 °C. After incubation, the results were determined based upon any color change or gas production in the MacConkey broth. The color change of the MacConkey broth from red to yellow and the gas production inside the Durham tube indicated a positive result for the presence of *E. coli* for the presumptive test.

If the broth color remained the same, meaning red and no gas formed, it showed a negative result for the presumptive test. Furthermore, a confirmatory test was carried out to ensure the presence of *E. coli* by testing the positive tubes of the presumptive test as many microorganisms are also present in water which can give a false presumptive test result. A confirmatory test was conducted to determine the presence of *E. coli* in the drinking water. This can be performed using either Brilliant Green Lactose bile Broth (BGLB) or Eosin Methylene Blue agar medium (EMB).

A positive presumptive test in the BGLB was used. A solution of brilliant green lactose broth medium was prepared using 10 g of peptone, 10 g of lactose, 20 g of bile salt, 0.0133 g of brilliant green and 1 L of distilled water. We sterilized the BGLB medium in an autoclave for 15 min at 121 °C.

We gently shook the positive presumptive tubes and then transferred a loop full of culture into the BGLB fermentation tube.

The brilliant green dye in the BGLB medium inhibited the growth of Gram-positive bacteria. The test tubes were then incubated for 48 h at 35 °C, and were observed to contain BGLB medium and inoculum of the positive presumptive test for the production of gas in the inverted Durham table.

The production of gas in the BGLB medium confirms the presence of *E. coli* [2,36].

#### 2.6.4. Antibiotic Sensitivity Analysis

Antibiotic susceptibility tests were performed on all water samples collected and on all *E. coli* positive isolates using the disc diffusion method according to ISO standards [37,38].

The materials used were Mueller Hinton Agar, sterile Petri dishes, forceps, chemical balance, a conical flask, cotton wool, and a cotton swab (sterile), a ruler, an incubator, micro flow and antimicrobial agents.

The preparation of Mueller Hinton Agar encompassed suspending 38 g of the medium in one liter of distilled water. It was heated for frequent agitation and boiled for one minute to completely dissolve the medium.

It was then autoclaved at 121 °C for 15 min and cooled at room temperature.

We poured cool Mueller Hinton Agar into the sterile Petri dish on a horizontal level surface to give uniform depth. We allowed it to cool at room temperature.

From a pure bacterial culture, we took two or three colonies with a sterile cotton swab. We inoculated the agar by streaking with the swab containing the inoculum. We allowed the surface of the medium to dry so as to create absorption of excess moisture.

Using sterile forceps, we placed an antibiotic disc on the surface of the inoculated and dried plate. Immediately, we pressed it to ensure complete contact between the disc and the agar surface. We incubated the plates in an inverted position at 35 °C or at an optimum growth temperature.

Furthermore, we observed the inhibition zones after 18 to 24 h. The inhibition zones are the points at which no growth is visible to the unaided eye. We then used a ruler to measure the diameter zones and recorded the values obtained. We rounded up the zone measurement to the nearest millimeter. It showed the bacteria to be either resistant (R) or susceptible (S) depending on the size of the zone. The antibiotics or antimicrobial agents used and their concentrations are shown below in Table 1.

The discs were placed on agar plates and incubated for between 24 and 48 h. A counting chamber and magnifier was used to count the *E. coli* colonies and the amount of *E. coli* grown on a culture per 100 mL [39].

The primary data sources were the laboratory results registers at the Milton Margai Technical University.

### 2.7. Assessing the Physical Structures of Water Sources

A WHO-recommended checklist [40] was adapted to systematically assess the physical structures of the fifteen standpipes (see Box 1) and five community wells (see Table 2) using a paper-based approach. This checklist was pretested before use. It was used to assess if the standpipe or well was likely to be contributing to the risk of contamination while people were collecting water. The presence or absence of latrines within 30 m of a well or standpipe was used to determine whether the wells or standpipes met the requirements of the technical design and specification manuals. The physical structures of the standpipes and wells were observed, photos were taken, and users were interviewed on safe collection and transportation practices.

Box 1Questions used to assess the condition of standpipes in Brookfields and Wilberforce communities in Freetown Sierra Leone in May 2021 (1 is a positive finding, 0 is negative).1. Is the tap functioning and not physically damaged?2. Is the standpipe placed properly?3. Is the tap in good condition (not leaking)?4. Is there no latrine within 30 meters distance?5. Is the drainage channel well maintained and not cracked, broken or in need of cleaning?6. Is the fence complete and not missing or faulty?7. Does any spilt water drain away without collecting in the drainage area?8. Is the concrete floor in good condition and not cracked?9. Is the standpipe tap securely attached to the head?10. Is the tap-cover secure?

### 2.8. Analysis and Statistics

The collected data were entered into the EpiData Entry software (v 3.1, EpiData Association, Odense, Denmark). Data validation was carried out in a systematic way to minimize errors or omissions in the reported laboratory results. Frequencies, proportions, and means were described and analyzed. Differences in proportions between the first phase and second phase data collections, the geographic location, and whether the taps were on the pipe network or attached to a storage tank were compared using the chi square test and *p*-values. Differences at the 5% level (*p* < 0.05) were regarded as significant.

### 2.9. Institutional Review Board Statement

Ethics approval was obtained from the Sierra Leone Ethics and Scientific Review Committee (SLESRC) of the Ministry of Health and Sanitation, Government of Sierra Leone on 5 July 2021) and the International Union against Tuberculosis and Lung Disease Ethics Advisory Group, Paris, France (EAG number 06/21). 

## 3. Results

### 3.1. Physical Assessment of Wells and Standpipes

All the wells were poorly set up and badly maintained. The mean score was 2.4 out of 12. Upon inspection, there were many issues (see Table 3), with poor fencing, the potential for underground seepage of latrines into the wells, and poor drainage of water from around the well. No well had a hand pump but they all had a bucket to lower into the well on a rope, which was the only positive finding common to all the wells.

Using the questions in Table 1 for the inspection, the standpipes scored between 7 and 10 out of 10, with a mean of 8.8.

At nine standpipes (60%), spilt water did not drain away without collecting near the standpipe. At six standpipes (40%), the surrounding fence was damaged. The drainage channel was damaged in two taps (13% of standpipes), and with one standpipe, a latrine was less than 30 m away.

There was no change in the scores between the two rounds of inspections in the dry season and rainy season for either the standpipes or wells, apart from one standpipe which had been badly damaged and was no longer functional.

### 3.2. Physicochemical and Microbiological Parameters

Water samples were taken from each well and standpipe and sent for laboratory analysis.

The samples from the five wells all showed poor quality water with raised levels of TDS between 33 and 70 NTU (recommended level <10 NTU), all wells had at least a low level of risk from *E. coli* (one to 10 *E. coli* per 100 mL) and one well had a very high level of risk (1600 *E. coli* per 100 mL); three of the wells had elevated nitrate levels between 13 and 15 parts per million (recommended <10 ppm).

The samples taken from the water company treatment plant showed that, as expected, the water quality improved as it progressed through the treatment plant and that when it was discharged from the treatment plant, the turbidity was at an acceptable level (5 NTU). The *E. coli* count fell from 17 (95% CI 7–40) at the water source to 2 (95% CI 0–4) at discharge, so was still contaminated with *E. coli* at discharge. WHO defines low risk as 1–10 per 100 mL sample using the MPN method. The pH was within acceptable levels and ranged from 6.0 to 7.5 for all samples tested.

All the samples from standpipes were assessed for the presence of *E. coli* using the MPN method (Table 3). All these water samples were contaminated with *E. coli* at least at a low risk level (1–10 MPN/100 mL) of *E. coli*. The proportion of the samples that contained *E. coli* at an intermediate risk level (10–100 MPN/100 mL) or above varied from 20% to 33% (see Figure 3), although the standpipes connected to the network pipes did not show any more than a low risk of *E. coli*. The number of standpipes directly connected to the pipe network was too small a sample to confirm the significance of this finding. There was no association between finding *E. coli* in a sample and the geographical location or season of the year.

The pH of all the samples from the standpipes was in the range of pH 6.0 to 7.5, which is within acceptable levels as indicated in the WHO drinking water quality guidelines.

The proportion of samples with an elevated level of TDS (above 10 NTU) varied (Table 4. There was a statistically significant (*p* < 0.0001) association between the season and level of turbidity, with a higher proportion of samples being contaminated in the rainy season (73% vs. 7%). There was no association between level of turbidity, the locality or how the standpipe received its water.

The proportion of samples with elevated nitrate levels (10 parts per million) in the water varied from 13% to 21% (Table 5), except for water from the standpipes connected to the network where there were no elevated nitrate levels. Again, this was too small a sample to test for statistical significance. All the samples were tested for lead, zinc and phosphate, and all were below 10 parts per million and posed no risk.

### 3.3. Culture and Antibiotic Sensitivity Testing

All the 48 water samples collected were cultured. Of these, 25 (52%) grew *E. coli*. Antibiotic sensitivity testing was performed on all the *E. coli* isolates using the disc diffusion method for ampicillin, gentamicin, penicillin, chloramphenicol and sulphafuzole. No antimicrobial resistance was shown in any of the isolates.

### 3.4. Examples and Study Photos

One of the ways drinking water sources from wells are contaminated are shown in Figure 4. Drinking water were tested for pH, and TDS connected to a 10,000 L capacity standpipe (see Figure 5), and Figure 6 depicted a standpipe connected to a pipe network.

## 4. Discussion

This study confirms published research conducted in Bo, Sierra Leone and elsewhere showing that wells are more susceptible to contamination than standpipes [41,42]. In our study, all five wells were contaminated with *E. coli.* The water quality in the standpipes was lower than desirable as all samples from standpipes contained at least a low risk level from *E. coli,* and around 25% of standpipe samples had at least an intermediate risk level of *E. coli*. Water tested at discharge from the water company treatment plant was contaminated with *E. coli* at a low risk level.

The presence of *E. coli* in drinking water above the recommended WHO drinking water quality guidelines (Zero (0) *E. coli* per 100 counts per mL) is a cause for concern as it suggests fecal contamination with the risk of transmission of many diseases such as typhoid, paratyphoid, diarrhea, and cholera.

The nitrate concentrations exceeded the WHO-recommended levels for safe drinking water in 60% of the wells and in less than 20% of the standpipes. This may reflect contamination from agricultural fertilizers or human or animal waste. The elevated levels of nitrates bring health risks due to chemical contamination of drinking water, which include skin lesions, vascular and cardiac problems, and cancer of the bladder, lungs and birth defects [43].

Other studies conducted in China and elsewhere [8,9,10,13] frequently found antibiotic resistant *E. coli* in drinking water sources, but our study found no antibiotic resistance with common antibiotics such as ampicillin, gentamicin, penicillin, chloramphenicol and sulphafuzole. This is reassuring but will require ongoing monitoring to see if this situation changes. Other studies conducted in Sierra Leone have also found that wells were more contaminated with *E. coli* than standpipes [41,42].

This research confirms the findings of the 2017 Multiple Indicator Cluster Survey [8] that drinking water in Freetown is contaminated with *E. coli*. They found that 88% of the household population in the Western Area had *E. coli* in their drinking water source, whereas we found 100% of samples had at least a low level of *E. coli* and 20–33% of samples were contaminated with *E. coli* at an intermediate level. It appears the situation is worse now than in 2017 and it contravenes the WHO drinking water quality guideline of no *E. coli* being present in drinking water. Our findings that wells are sources of poorer quality drinking water compared to the pipe water system is not a surprise. The drinking water quality needs improvement in these areas of Freetown.

Wells are more susceptible to contamination because of underground seepage from latrines within 30 m distance. They have not been constructed or rehabilitated using the recommended guidelines used nationally for technical design [44].

Water tested on discharge from the water company treatment plant remained contaminated at a low risk level for *E. coli* and around 75% of standpipe samples also contained a similar level. It is therefore unclear whether the source of this contamination is from sources near the standpipe, in the water distribution network or as a result of the quality of water discharged from the treatment plant.

The standpipes were directly connected to a pipe network or to storage tanks. The six samples from the three standpipes connected directly to the pipe network did not display any *E. coli* growth. It appears that standpipes being connected to a storage tank may have created additional potential sources of contamination, such as in the bowser during transportation of water from the treatment plant to the different locations or when water is transferred from the bowser to the storage tank. This needs further research.

The inhabitants rely on the water treatment carried out by the public water company. The turbid water in the standpipes may be due storage tanks not being cleaned and chlorinated on a regular basis to ensure the water is safe for consumption. Furthermore, some of the standpipes with tanks may be at risk of further contamination from mud around the base of the taps because the taps were not placed on a concrete floor.

Our study has several strengths. All the sampled sites were visited and inspected using a standard WHO-adapted checklist. Samples were also collected from the field and transported to the MMTU using recognized procedures. The samples were tested for physicochemical, microbiological, and antibiotic susceptibility at the university laboratory in Freetown which had the capacity and competency to perform such work.

A limitation of this study was that we only targeted two communities in Freetown and we are uncertain how representative the findings are across the city. Given the lack of difference in the results from the two areas, the findings may be applicable to other areas in Freetown and possibly more widely. We had initially planned to use ten wells and ten standpipes in our study. However, we found that there were markedly fewer wells than expected. We were unable to collect samples from the bowser transporting water in tanks to the standpipes. We had too small a sample to determine whether there was any statistical significance between standpipes connected to the pipe network and those connected to a storage tank. Our study only measured water contamination at the water sources, not in the household, where the storage and treatment could have contributed to contamination before the point of use.

Contamination and pollution of drinking water has both public health and economic implications. The impacts may include an increase in human deaths, excessive health expenditures, loss of workdays, and a reduction in agricultural output and fish populations [43].

Our results suggest two key changes that could reduce the risk of *E. coli* contamination: Replace un-improved wells with standpipes;Monitor water points to ensure that basic standards of water quality treatment and cleaning are followed.

We realize that replacing all the wells may not be possible in the near future. Until then, other steps can be taken to reduce health risks. Communities and local public health leaders can work together to improve the condition of wells at the source and follow the recommended technical designs and specifications for the construction of wells. Chlorination of the wells at the source can also be carried out to reduce the contamination.

Household water treatment methods such as filtration, sedimentation, boiling, ultraviolet radiation, use of chlorine compounds and safe storage can be used after collection of the water and before drinking.

The public water company should work with stakeholders to ensure close monitoring and surveillance of all the standpipes. Regular chlorine treatment of water supplied by the water company should be carried out by the staff of the water company. The standpipes connected to water tanks should be placed on concrete floors in order to avoid possible damage and cleaned according to an agreed schedule by the caretakers.

The risks from using these water sources could be reduced through awareness campaigns to engage community members and to increase knowledge and calls for improved water sources for both wells and standpipes. Water safety plans can be prepared and implemented for the wells and standpipes in those two communities to ensure protection of their drinking water. The high levels of TDS can be reduced for standpipes and wells using distillation, deionization, and reverse osmosis filters, all of which make drinking water safer and more acceptable for consumption.

It is important to investigate if the apparent lower water quality in standpipes attached to water tanks compared to those attached to the pipe network is a true finding. If it is, then steps to improve the water quality in the storage tanks should be undertaken. Further research is also necessary to investigate how household use, storage, and treatment could contribute to water contamination.

Regular household water quality monitoring and surveillance at point of use should be conducted by the MoHS in the two communities studied. Due to inadequate water infrastructure in these communities, capital investment and fresh funding are required to construct wells and standpipes.

## 5. Conclusions

This study has shown that the wells in these two communities of Freetown, Sierra Leone, are much more likely to be contaminated with *E. coli* and/or have elevated TDS than standpipes. In addition, once the water has left the water company treatment plant, there is an appreciable risk of it being contaminated again with *E. coli* or TDS before it is collected at a standpipe. Identifying where this contamination is occurring requires further research. Where *E. coli* was grown from the water samples, no antimicrobial resistance was found.

## Figures and Tables

**Figure 1 ijerph-19-06650-f001:**
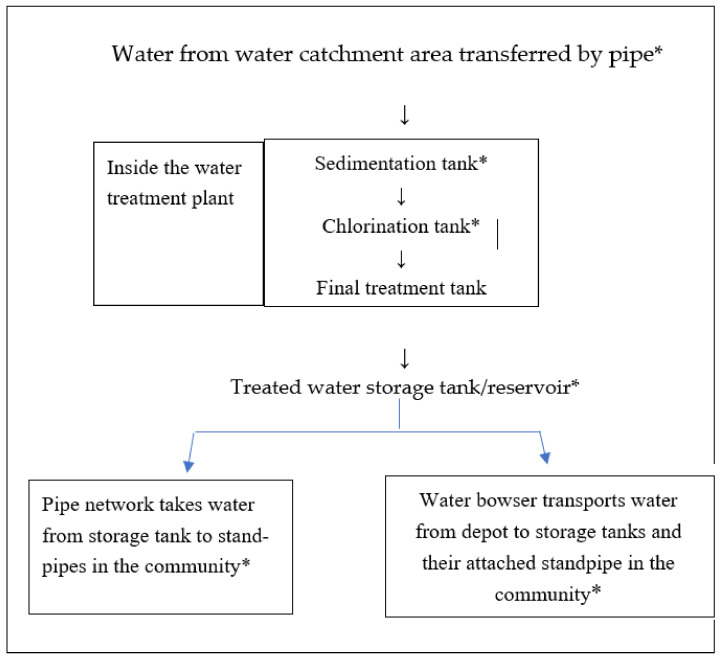
Flow diagram of water travelling from the catchment area of public water company to standpipes in Freetown Sierra Leone and the stages at which the water samples were taken. * indicates where water samples taken.

**Figure 2 ijerph-19-06650-f002:**
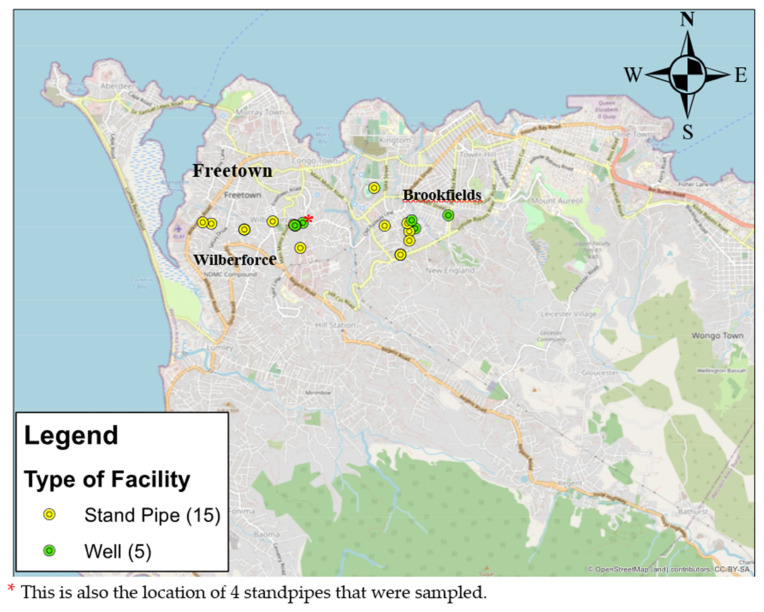
Map of Freetown indicating the sampling sites of the wells and standpipes in Western Area, Sierra Leone.

**Figure 3 ijerph-19-06650-f003:**
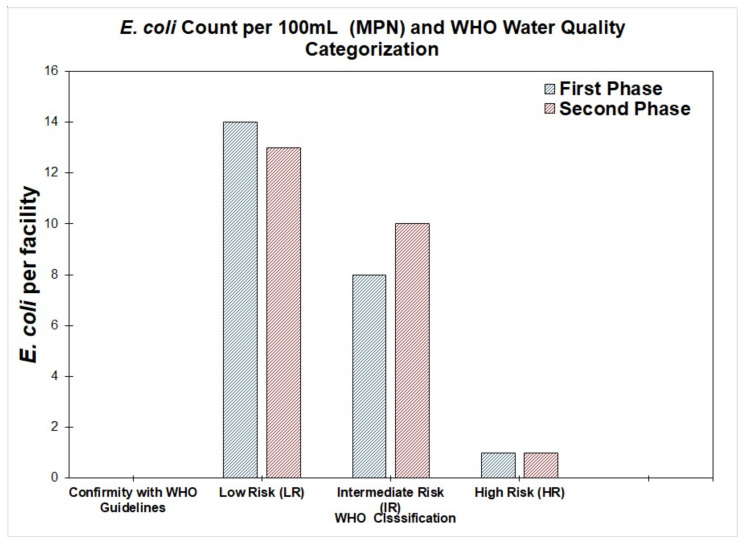
*E. coli* count per 100 mL according to the WHO standard *E. coli* classification for the first and second phase sample collections.

**Figure 4 ijerph-19-06650-f004:**
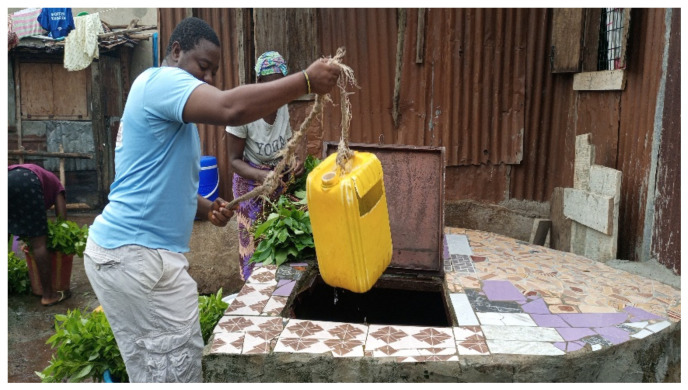
Sources of contamination at a well. Leafy vegetables are washed on top of the well, and that same water may drain back into the well. While the woman washes the vegetables, the man fetches water using a plastic container hung from a rope. The lid or cover is also prone to corrosion as it is made of iron and can be another potential source of contamination.

**Figure 5 ijerph-19-06650-f005:**
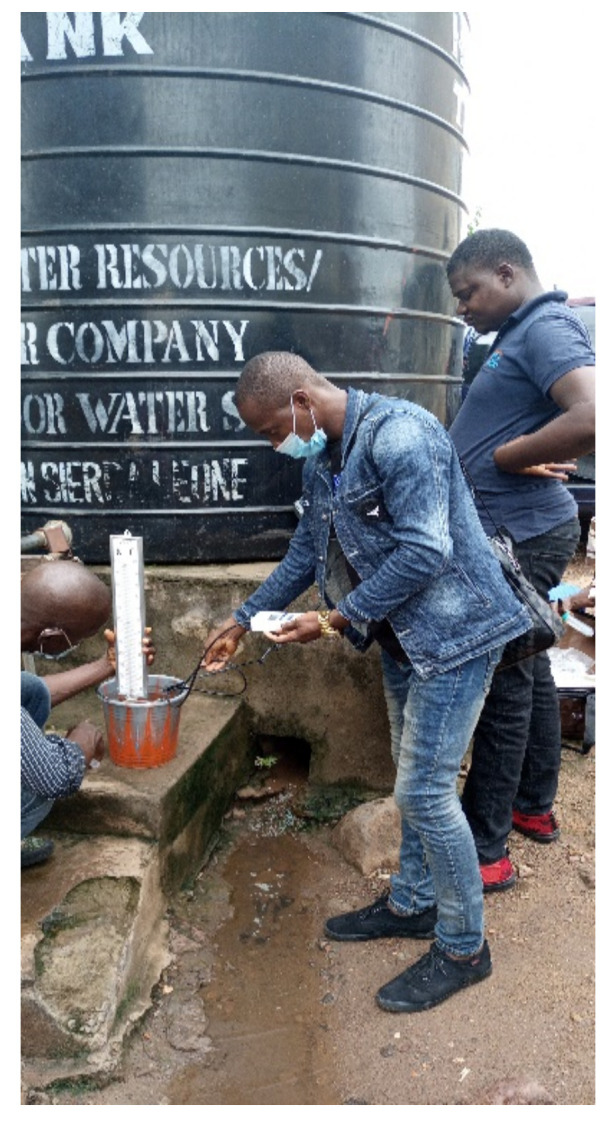
Physical testing of a standpipe connected to a storage tank. A standpipe connected to a storage tank (10,000 L capacity) depicts physical testing of the pH of the water and total dissolved solids using pH meter and turbidimeters.

**Figure 6 ijerph-19-06650-f006:**
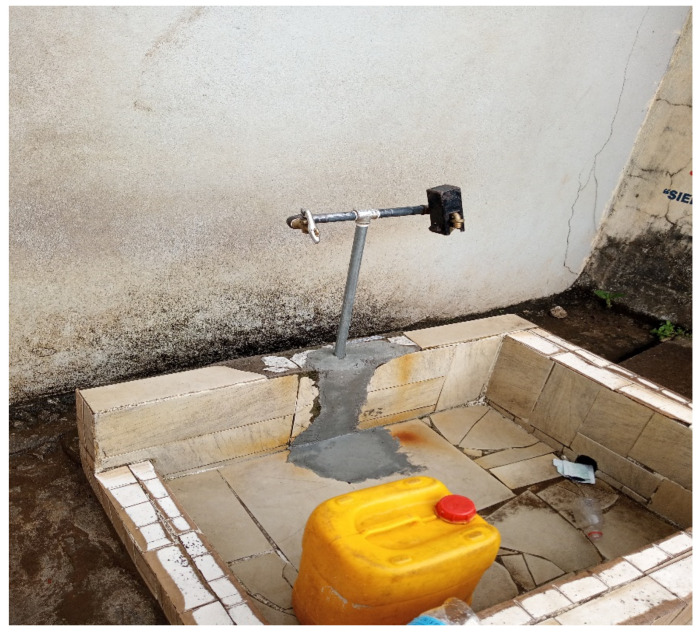
A typical example of one of the standpipes in Brookfields connected to the pipe network. The tap is closed after use by the people.

**Table 1 ijerph-19-06650-t001:** Antibiotic discs and concentration used for antimicrobial resistance testing of water samples taken in Freetown Sierra Leone.

Antibiotics	Disc—Microgram (µg) Concentration (Content)
Ampicillin	2 µg
Chloramphenicol	10 µg
Erythromycin	10 µg
Penicillin	1.5 µg
Sulphafuzole	100 µg
Gentamicin	5 µg

**Table 2 ijerph-19-06650-t002:** Questions used to assess the condition of wells in Brookfields and Wilberforce in Freetown Sierra Leone in May 2021 (1 is a positive finding, 0 is negative).

Question	Well 1	Well 2	Well 3	Well 4	Well 5	Row Total
1. Is the drainage channel complete and not cracked, or in need of cleaning?	0	0	0	0	0	0
2. Is the well-cover securely attached?	0	0	0	0	0	0
3. Spilt water drains away and does not collect in the apron area?	0	0	0	0	0	0
4. Is there no latrine within 30 m distance?	0	0	0	0	0	0
5. Is the hand pump present?	0	0	0	0	0	0
6. Is the concrete apron correctly fitted and not cracked around the well?	1	0	0	0	0	1
7. Is the fence complete and not missing or faulty?	0	0	1	0	0	1
8. Is the head wall neatly dressed and in good condition?	0	0	1	0	0	1
9. Is the cover slab well maintained?	1	0	1	0	0	2
10. Is the slab placed properly to protect the well?	1	0	1	1	0	3
11. Does the drainage prevent pooling of water within 2 m of the well?	1	1	1	0	0	3
12. If there is no pump, is there a bucket attached to a rope for lowering into the water?	1	1	1	1	1	5
Score out of 12	5	2	6	2	1	

1 = yes, 0 = no

**Table 3 ijerph-19-06650-t003:** Number of samples with *E. coli* present at an intermediate risk level or above, using the MPN method values in water from standpipes in Freetown, Sierra Leone.

Variable	Total Samples	*E. Coli* Present in Sample (%) ^a^		*p*-Value
Brookfield area	14	3	(21)	
Wilberforce area	16	4	(25)	0.41
Dry season	15	4	(27)	
Wet season	15	3	(20)	0.33
Standpipe on network	6	0	(0)	
Standpipe on tank	24	7	(33)	0.13 *

* Fisher’s exact test. ^a^ Percentages are calculated relative to the total samples that have *E. coli* present.

**Table 4 ijerph-19-06650-t004:** Number of samples with the total dissolved solids above the recommended level in water from standpipes in Freetown, Sierra Leone.

Variable	Total Samples	Number of Samples >10 NTU ^†^ (%) ^a^		*p*-Value
Brookfield	14	4	(29)	
Wilberforce	16	7	(44)	0.19
Dry season	15	1	(7)	
Wet season	15	11	(73)	<0.00001
Standpipe on network	6	3	(50)	
Standpipe on tank	24	8	(33)	0.22

^†^ Nephelometric Turbidity Units. ^a^ Percentages are calculated relative to the total samples with >10 NTU. Standard range used: >10 NTU.

**Table 5 ijerph-19-06650-t005:** Number of samples where the nitrates were above the recommended level in samples from standpipes in Freetown Sierra Leone.

Variable	Total	Number of Samples with Nitrates above Recommended Level (%) ^a^		*p*-Value
Brookfield	14	2	(14)	
Wilberforce	16	3	(19)	0.37
Dry season	15	3	(20)	
Wet season	15	2	(13)	0.31
Standpipe on network	6	0	(0)	
Standpipe on tank	24	5	(21)	0.30 *

* Fisher’s exact test. ^a^ Percentages are calculated relative to the samples of nitrates above the recommended level (10 parts per million) in the sample. Standard range used: >10 parts per million.

## Data Availability

The data for this study are available at the Ministry of Health and Sanitation repository and will be shared with the Editorial team upon request. The metadata record of the data used in this paper is available at DOI-10.6084/m9.figshare.19205664. Requests to access these data should be sent to the corresponding author.

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
