# Peer review of "Evaluation of Drinking Water Quality and Bacterial Antibiotic Sensitivity in Wells and Standpipes at Household Water Points in Freetown, Sierra Leone"

_ijerph, 2022, doi:10.3390/ijerph19116650_

Round 1

Reviewer 1 Report

Manuscript. Number: 1634874

Overall, a lot of additional work is needed in introduction and methods to make the data currently displayed in the manuscript more comprehensible and interpretable by the readers of this journal. I recommend the authors provide a map of the study with sampling sites on it.

Specifically, please see my suggestion below based on the individual section of the manuscript:

INTRODUCTION:

Overall, the entire introduction needs a lot more work and also lacks citations. A lot more citations are required throughout the introduction to substantiate the importance of this work. For something like this which has a global significance (paras 1 and 2), statements should not be supported by just a citation or two. Why was E. coli used as an indicator? What is the significance of measuring microbial contamination using E.coli (USEPA standards should be mentioned here). Why was antibiotic resistance tested?

METHODS:

  • Para 1 and 2 under methods should be in introduction.
  • E. coli needs to be corrected for proper scientific writing throughout the manuscript.
  • Section 2.3: suggesting inclusion of a study area map consisting of the locations sampled clearly pointed out.
  • Section 2.6: All of the assays in the study needs to be described in much more detail. How were the spectrophotometric measurements for each parameter conducted? Authors need to clearly mention the methods here, a citation is needed to provide information on where the methods were taken from, but just a citation is not enough. However, here a citation is also not provided.
  • How were the coli MPNs measured? – need detailed methods here, together with proper citation.
  • How were the antibiotic resistance analysis done? What were the antibiotics? What were their concentrations? What method was followed? Where is that citation?

RESULTS:

Section 3.1 is not necessary.

I suggest a plot or figure for the E. coli numbers. Authors also need to clearly mention what the standard ranges are and where those are referenced from.

Author Response

Thank you very much for the email I received on the 25th of March, 2022 with feedback on the manuscript we submitted. We have carefully gone through the comments and suggestions you made and have revised our paper accordingly. We feel that the paper is much improved as a result of this peer review process, and thank you for taking it to this stage.

Please find for your kind consideration the following:

  1. A “point by point” response to the comments and suggestions of the reviewers (below pages 2-5).
  2. A new revised version of the manuscript marked R1. We have used track changes in order to facilitate review.

Reviewer 2 Report

This paper looks at the contamination of water sources in Freetown, Sierra Leone.

An explanation of standpipe v well should be added to the introduction.

Esherichia Coli should be Escherichia coli and E. Coli should be E. coli in all instances.

Please expand greatly on the study design so that some understanding can be gained.

Figure 1 is unreadable in its current form 

Please expand on how the  "levels of nitrate, lead, zinc and phosphate" were determined.

How were E.coli cultured? what specifically was done to determine they were E.coli  and not just coliforms? 

"Antibiotic susceptibility tests were performed on all water samples collected and on all E. Coli positive isolates by the disc diffusion method"

What Antibiotics were used (This should be in Materials and Methods)? What breakpoints (EUCAST? CSLI) were used to determine susceptibility or resistance? What media was used? How were samples were prepared? 

Reference 10 The water company website itself should be referenced

Author Response

Thank you very much for the email I received on the 25th of March, 2022 with feedback on the manuscript we submitted. We have carefully gone through the comments and suggestions you highlighted and have revised our paper accordingly.

We feel that the paper is much improved as a result of this peer review process, and thank you for taking it to this stage.

Please find for your kind consideration the following:

  1. A “point by point” response to the comments and suggestions of the reviewers (below pages 2-5).
  2. A new revised version of the manuscript marked R1. We have used track changes in order to facilitate review.

Round 2

Reviewer 1 Report

Authors have made all corrections based off my previous suggestions. The manuscript now reads much better.

One quick suggestion: Table 1: for microgram, instead of U, you can use the micro symbol from MS Word. 

Author Response

Assistant Editor

MDPI

25th April 2022

RE: IJERPH-1634874

Dear Dr. Nikolina Trninic,

Thank you very much for the email I received on the 21st of April, 2022 with feedback on minor revision for the manuscript we submitted. We have carefully gone through the comments and suggestions from the reviewers and have revised our paper accordingly. We feel that the paper is much improved as a result of this peer review process, and thank you for taking it to this stage.

Please find for your kind consideration the following:

  1. A “point by point” response to the comments and suggestions of the reviewers (below pages 2-5).
  2. A new revised version of the manuscript marked R1. We have used track changes in order to facilitate review.

While hoping that these changes would meet with your favourable consideration, we are willing to respond with any further information or other changes you might require.

Yours sincerely

Mr. Dauda Kamara.

(Corresponding Author)
